# The evolution of the temporal program of genome replication

Nicolas Agier[1], Stéphane Delmas[1], Qing Zhang [1], Aubin Fleiss[1], Yan Jaszczyszyn[2], Erwin van Dijk[2], Claude Thermes[2], Martin Weigt [1], Marco Cosentino-Lagomarsino [1,3] & Gilles Fischer [1]

Genome replication is highly regulated in time and space, but the rules governing the remodeling of these programs during evolution remain largely unknown. We generated genome-wide replication timing profiles for ten *Lachancea* yeasts, covering a continuous evolutionary range from closely related to more divergent species. We show that replication programs primarily evolve through a highly dynamic evolutionary renewal of the cohort of active replication origins. We found that gained origins appear with low activity yet become more efficient and fire earlier as they evolutionarily age. By contrast, origins that are lost comprise the complete range of firing strength. Additionally, they preferentially occur in close vicinity to strong origins. Interestingly, despite high evolutionary turnover, active replication origins remain regularly spaced along chromosomes in all species, suggesting that origin distribution is optimized to limit large inter-origin intervals. We propose a model on the evolutionary birth, death, and conservation of active replication origins.

---

[1] Sorbonne Université, CNRS, Institut de Biologie Paris-Seine, Laboratory of Computational and Quantitative Biology, F-75005 Paris, France. [2] Institute for Integrative Biology of the Cell, UMR9198, CNRS CEA Univ Paris-Sud, Université Paris-Saclay, 91190 Gif sur Yvette Cedex, France. [3] IFOM (FIRC Institute of Molecular Oncology), Via Adamello 16, 20139 Milano MI, Italy. Correspondence and requests for materials should be addressed to G.F. (email: gilles.fischer@sorbonne-universite.fr)

To ensure the completion of genome doubling before cell division, eukaryotic chromosomes initiate DNA replication from multiple sites, termed replication origins. Mammalian and yeast genomes initiate replication at hundreds and thousands of active origins respectively, that are selected from a larger pool of possible autonomously replicating sequences (ARS)[1–3]. In yeast, active origins are distributed throughout the genome, mainly at non-transcribed and nucleosome-depleted sequences and comprise a specific DNA motif called ARS consensus sequence (ACS) which is bound by the Origin Recognition Complex[4–6]. Yet only a subset of the pre-determined replication initiation loci will be activated by different cells in a population. DNA combing and fluorescence microscopy confirmed the stochastic activation of origins at the individual cell level[7–10]. Nevertheless, within populations, a specific temporal program of genome replication emerges and can be recapitulated by averaging the heterogeneous replication kinetics of a large number of cells[7,11]. This can be achieved experimentally by time-course measurements of replication progression during S-phase in a cell population, using microarray hybridization or high throughput sequencing[11]. Mathematical modeling of such replication timing data gives access to the stochastic firing components of individual origins[10,12–14]. First, the efficiency describes the probability of activation for each replication origin, which represents the proportion of cells in a population in which the origin actively fires[15,16]. Second, each origin has an intrinsic strength called characteristic firing time, which represent its time, early to late, of activation during S-phase in the absence of interfering neighboring origins[17,18]. Firing time and efficiency both reflect the probability of origin firing, either per unit of time or over the entire S-phase respectively. Consequently, timing and efficiency are correlated[16,19,20], the fundamental difference being that efficiency incorporates the effect of passive replication from forks originating from different origins while firing rate is an origin-specific quantity.

Early microarray-based studies in *Saccharomyces cerevisiae* established the first genome-wide replication program providing both replication timing information for all genomic sequences and replication origin location along chromosomes[18,21]. Since then, temporal programs of genome replication have been established for many eukaryotic genomes including three closely related *Saccharomyces* sensu stricto species[22] and nine more distantly related species (*Candida glabrata, Naumovozyma castellii, Tetrapisispora blattae, Zygosaccharomyces rouxii, Kluyveromyces lactis, Lachancea waltii, Lachancea kluyveri, Pichia Pastoris* and *Candida albicans*)[3,23–27]. An early comparative genomics study among closely related *Saccharomyces* species showed that phylogenetic conservation can be used to determine the genome-wide location of replication origins in *S. cerevisiae*[2]. Additional comparative genomic approaches later confirmed the

existence of conserved sequence elements necessary for origin function in other yeast genomes as well as a conserved role for centromeres and telomeres in defining early and late origin firing, respectively[25,26,28–30]. Comparative analysis of replication timing also revealed that at short evolutionary distances, between *Saccharomyces* species, most active origins remained conserved both in location and in activation time, resulting in an important conservation in the temporal order of genome replication[22]. On the contrary, comparisons between more distantly related species revealed that the conservation of the temporal organization of replication was restricted to specific genetic elements such as centromeres and histone genes that are among the first regions to replicate and telomeres that are among the last[27]. Only a small proportion of replication origins (5−30%) are conserved in position between *S. cerevisiae, L. waltii, L. kluyveri* or *K. lactis*[3,23,28]. Such a level of reprogramming contrasts with the global conservation observed between *Saccharomyces* genomes and precludes any chance to identify the selective forces responsible for the conservation, gain and loss of replication origins over evolutionary time.

To overcome these limitations, we focus on the continuous evolutionary range covered by the genus *Lachancea*, from closely related to more diverged species[31]. We first characterize the genome replication dynamics and origin usage at the population level in ten *Lachancea* species. This unique dataset allows us to infer all events of origin gains and losses since these species diverged from their last common ancestor. We then correlate the functional properties of replication origins from equivalent evolutionary ages, such as their chromosomal location, firing time and efficiency to reveal new rules that govern the birth, death and conservation of active replication origins during evolution.

## Results

**Temporal programs of genome replication in *Lachancea*.** We measured the temporal programs of genome replication by assessing DNA copy number change during S-phase, for ten *Lachancea* species with high-quality genome assemblies, in order to determine the mean replication time, called Trep[31–33] (Supplementary Fig. 1 and Supplementary Data 1). Cells were synchronized by physical discrimination of G1 cells and synchronously released into S-phase. Time-point samples were taken during S-phase until cells reached the G2 phase and DNA samples were analyzed using Illumina deep sequencing[34]. We also performed a Marker Frequency Analysis (MFA), consisting in measuring replication dynamics directly from an exponentially growing cell population[35] and compared our results to published profiles for *L. waltii* and *L. kluyveri*[3,23]. For each species, the Trep and MFA profiles were highly correlated, indicating good reproducibility (Supplementary Fig. 2). In addition, we inferred

**Table 1 Replication profile features**

| Species | Genome size | Nb ORI | ORI every ... | Median replication time (min) | | | Average S-phase duration (min) |
|---|---|---|---|---|---|---|---|
| | | | | Centromere | Telomere | Histone | |
| *L. fantastica* | 11.3 Mb | 256 | 44 kb | 6 ± 2 | 29 ± 7 | 3 ± 3 | 39 |
| *L. meyersii* | 11.3 Mb | 237 | 48 kb | 2 ± 2 | 17 ± 5 | 1 ± 1 | 24 |
| *L. dasiensis* | 10.7 Mb | 234 | 46 kb | 4 ± 2 | 24 ± 6 | 3 ± 1 | 36 |
| *L. nothofagi* | 11.3 Mb | 241 | 47 kb | 6 ± 4 | 28 ± 9 | 3 ± 1 | 42 |
| *L. waltii* | 10.2 Mb | 225 | 46 kb | 9 ± 3 | 24 ± 11 | 7 ± 1 | 41 |
| *L. thermotolerans* | 10.4 Mb | 211 | 49 kb | 2 ± 2 | 16 ± 5 | 1 ± 1 | 29 |
| *L. mirantina* | 10.1 Mb | 202 | 50 kb | 7 ± 3 | 35 ± 16 | 1 ± 1 | 48 |
| *L. fermentati* | 10.3 Mb | 214 | 48 kb | 2 ± 4 | 42 ± 4 | 1 ± 1 | 47 |
| *L. cidri* | 10.1 Mb | 200 | 51 kb | 4 ± 3 | 25 ± 4 | 2 ± 1 | 41 |
| *L. kluyveri* | 11.3 Mb | 244 | 46 kb | 1 ± 1 | 26 ± 6 | 1 ± 1 | 32 |

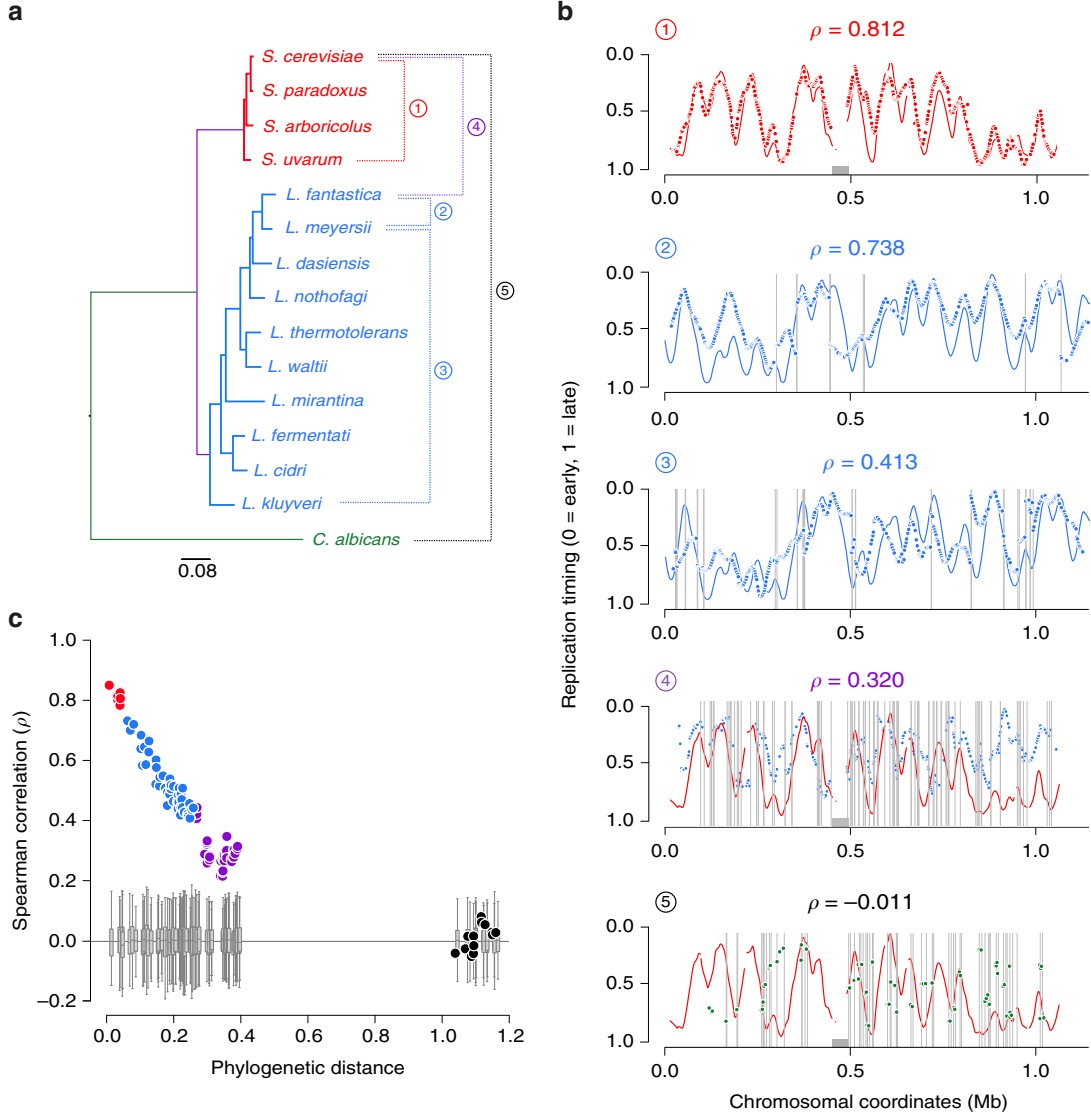

**Fig. 1** Evolution of replication timing profiles in yeast. **a** Phylogeny of 15 yeast species inferred from a maximum likelihood analysis of a concatenated alignment of 510 protein families. The circled numbers to the right of the tree relate to the pairwise species comparisons presented in **b**. **b** Synteny-based projections of replication timing profiles between pairs of species. We projected the replication timing of one species onto the chromosomal coordinates of a second species, using synteny conservation between the two genomes. The color coding is identical to **a**. The rho value above each profile indicates the genome-wide Spearman's rank correlation coefficient between the replication timing programs of the two compared species. Vertical gray bars indicate the position of synteny breakpoints between genomes. Comparisons 1, 4, and 5 represent the projections of *S. uvarum*, *L. fantastica*, and *C. albicans* timing data, respectively, onto the replication profile of *S. cerevisiae* chromosome XII. The gray rectangle symbolizes the rDNA locus in this chromosome. The comparisons 2 and 3 represent the projections of *L. fantastica* and *L. kluyveri* timing data, respectively, onto the replication profile of *L. meyersii* chromosome 0D. **c** Correlation between replication profile conservation and phylogenetic distance. Spearman coefficients on the *y*-axis correspond to the synteny-based projections of genome replication timing profiles between pairs of species as illustrated in **b**. Red, blue, purple, and black dot series correspond to intra-*Saccharomyces*, intra-*Lachancea*, *Saccharomyces* vs *Lachancea* and *C. albicans* vs all other species comparisons, respectively. The gray boxplots show the Spearman correlation values of a null model where, for each pairwise comparison, we applied a random offset to the coordinates of the syntenic genes in one of the two compared genomes. Whiskers are defined by selecting the values within the range of the 75th and 25th percentile plus or minus 1.5 × (75th percentile – 25th percentile) respectively, and at the maximum distance from the median. The offset was re-defined 100 times in both directions and correlations were calculated for each combination of one offset profile from one genome and one original profile from the other genome

the location of replication origins along the chromosomes by applying a peak calling method[34] and stringently defined active replication origins only if peaks were co-detected in both profiles (Supplementary Fig. 2 and Supplementary Data 1).

We identified 2264 active replication origins in the ten genomes (Supplementary Data 1). These genomes undoubtedly contain more weak origins with efficiencies too low to be detected by our approach. The total number of active origins per genome varies from 200 in *L. cidri* to 256 in *L. fantastica* (Table 1).

However, we observed a constant origin density along chromosomes of one origin every 47 kb (Supplementary Fig. 3a−c). Additionally, origins are more regularly spaced than what would be expected by chance (Supplementary Fig. 3d), as previously described for *S. cerevisiae*, *K. lactis*, *L. kluyveri*, and *L. waltii*[36]. We also found common features shared between all ten replication programs, such as the spatial alternation between early and late replicating large chromosomal regions, with the exception of the left arm of chromosome C in *L. kluyveri*[23,30], as well as both early

replicating centromeres and histone genes[27] and late replicating telomeres (Supplementary Fig. 1 and Table 1).

**Continuous evolutionary range of genome replication profiles.** We calculated the Spearman's rho for replication timing programs between pairs of species to determine their degree of conservation (see Methods). We included in the analysis our new *Lachancea* replication timing data and those previously published in *Saccharomyces* species[22], and *Candida albicans*[25] (Fig. 1a, b). We found high degrees of replication profile correlation between all sensu stricto *Saccharomyces* species, varying from 0.79 to 0.86, as previously reported[22]. On the contrary, all pairwise comparisons involving species from different genera (*Saccharomyces*, *Lachancea* or *Candida*) show little to no degree of correlation preventing any reliable comparative study. Interestingly, comparisons within the genus *Lachancea* revealed that the Spearman's rho stagger from highly conserved replication profiles ($\rho = 0.74$) to more variable programs ($\rho = 0.41$), with all intermediate levels of correlation in-between (Supplementary Fig. 4). These

coefficients linearly anti-correlate with phylogenetic distances ($R^2 = 0.88$, $P < 2.2 \times 10^{-16}$), showing that replication timing linearly evolves alongside protein divergence (Fig. 1c). This result indicates that the genus *Lachancea* is an ideal candidate to investigate the causes behind the progressive reprogramming of genome replication during evolution.

**Local impact of genome rearrangements on profile evolution.** We first determined that the number of syntenic homologs, considered as orthologous genes hereafter, used to project the replication timing of one genome onto another was globally constant, ruling out the possibility that the anti-correlation observed in Fig. 1c resulted from a bias in synteny detection (Fig. 2a). By contrast, the number of synteny blocks increases with phylogenetic distance (Fig. 2a) due to the accumulation of genome rearrangements during evolution[31,37,38]. Importantly, the anticorrelation slopes based on the local coefficients both near breakpoints and within conserved synteny blocks are similar to that of the global trend, revealing that the local effects of genome

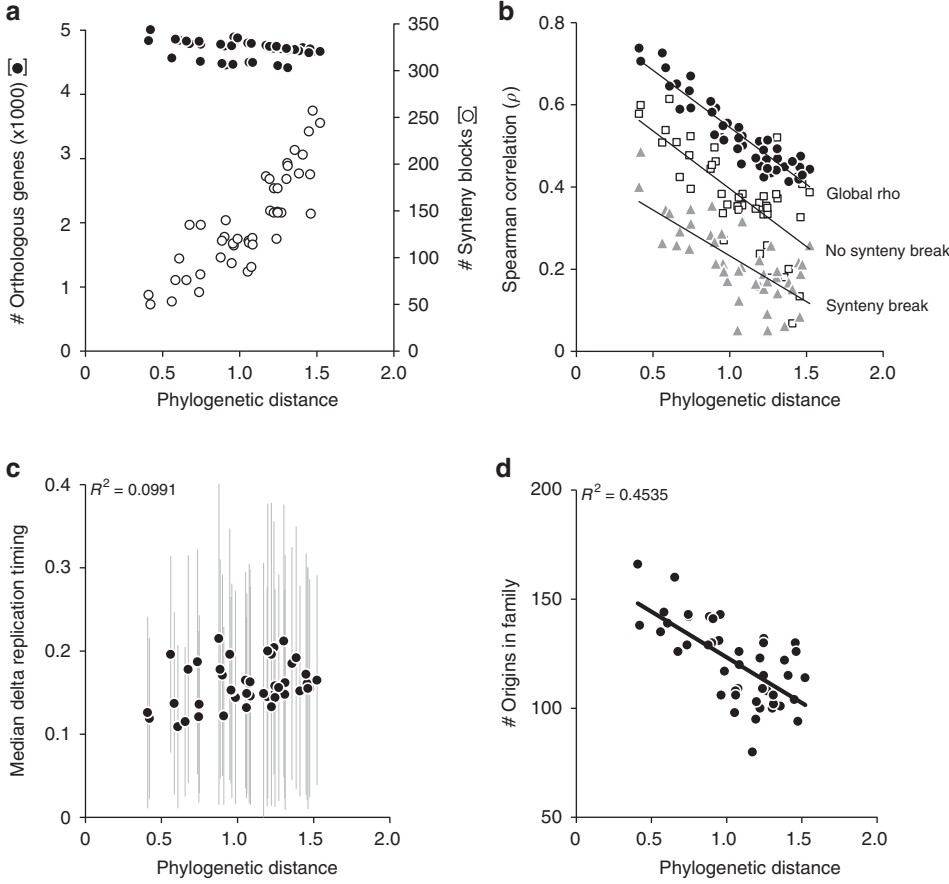

**Fig. 2** Evolution of genomic properties in the genus *Lachancea*. In all panels, each point corresponds to a pairwise comparison between two *Lachancea* species. The values of the property indicated on the y-axis are plotted as a function of the phylogenetic divergence between all pairs of *Lachancea* species as defined in ref. [31]. **a** The number of syntenic homologs and synteny blocks are represented by black and open circles, respectively. **b** Comparison of global and local correlation coefficients between pairs of genome replication timing profiles. The global coefficient plot (black circles) is identical to that of the blue dots in Fig. 1c ($R^2 = 0.868$, $P < 2.2 \times 10^{-16}$). Open squares and gray triangles represent the local correlation coefficients averaged over the same number of regions of five genes that are either located 25 genes away from the synteny breakpoints ($R^2 = 0.502$, $P = 2.24 \times 10^{-6}$) or directly flank the breakpoints ($R^2 = 0.477$, $P = 1.6 \times 10^{-6}$), respectively. **c** Origin firing time differences between conserved replication origins in all pairs of species. The y-axis corresponds to the normalized replication timing difference between orthologous origins. For each species, the average replication timing data (presented in Supplementary Fig. 1) were normalized between 0 and 1. The distributions of differences in normalized activation timing between orthologous origins are shown for all pairwise comparisons. The black dots indicate the median values for pairwise comparisons and the gray bars show the standard deviations. **d** Relative conservation of active replication origins in the different pairs of *Lachancea* genomes ($R^2 = 0.45$, $P = 3.9 \times 10^{-07}$). The y-axis corresponds to the number of active origins found to be conserved between each pair of species

rearrangements are not sufficient to explain the evolution of the replication programs (Fig. 2b). However, local Spearman's rho were on average weaker around synteny breakpoints than within conserved synteny regions, suggesting that genome rearrangements have a local impact on the evolution of replication profiles (Fig. 2b). Yet, part of this decay could be due to a technical component because projected profiles right at the edge of synteny blocks are necessarily discontinuous. Furthermore, if this was the sole component responsible for the correlation decay, the coefficients should increase abruptly in the next windows to reach the level found in conserved synteny regions. However, we found a gradual increase of the correlation coefficients with increasing distances from the breakpoints in the range of 5−20 genes indicating the implication of a biological component in the observed discontinuity of the profile at breakpoints (Supplementary Fig. 5). This biological component could be the consequence of fusing regions with different replication timing.

**Turnover of active origins drives timing program evolution.** We next sought whether the number of replication origins, their differential firing times, or their conservation levels could have driven the evolution of replication timing programs. First, we found no correlation between replication profile conservation and the raw number of active origins in the genomes (Supplementary Fig. 6). Second, we constructed families of orthologous replication origins based on synteny conservation (see Methods), to test whether replication program evolution could have resulted from differential conservation of active replication origins. After filtering 96 subtelomeric origins, we obtained 374 multi-species origin families comprising 1956 origins and 212 species-specific singleton origins (Supplementary Figs. 7 and 8). We found that the differences in origin activity, measured by peak height on replication timing profiles, between orthologous origins conserved in different species did not correlate with their phylogenetic distances. The range of median timing differences between orthologous origins remains between 10 and 22% despite varying phylogenetic distances between compared species (Fig. 2c). This absence of correlation suggests that the evolution of the

replication programs did not result from a progressive change in origin activation times. However, we found a negative correlation between the number of conserved replication origins within pairs of species and their phylogenetic distances (Fig. 2d). Given that phylogenetic distances also correlate with replication timing Spearman's rho (Fig. 1c), it results in the numbers of conserved origins correlating with the conservation of the replication timing profiles ($R^2 = 0.57$, $P = 2.5 \times 10^{-09}$). These results suggest that the appearance and disappearance of active replication origins would be the dominant process for shaping replication profiles during evolution.

**Dynamic evolutionary turnover of active replication origins.** We reconstructed the evolutionary history of active replication origins along the branches of the phylogenetic tree under a birth-death evolutionary model using *L. kluyveri* as the outgroup species (see Methods). We identified 1310 origins clustered in 220 families in the nine other *Lachancea* genomes that were vertically inherited from the last common ancestor of the clade designated as *L.A2* (Fig. 3a). We will refer to them as ancestral origins hereafter. Extant genomes comprise on average 68% of ancestral origins (Fig. 3b), varying from 59 to 83% (Supplementary Fig. 9). Only 37 out of these 220 ancestral families were faithfully transmitted from *L.A2* without any subsequent loss of origins along the branches of the tree. This means that 83% of the ancestral families underwent at least one event of origin gain or loss, demonstrating that the evolutionary turnover of the cohort of active replication origins is very dynamic (Fig. 3b). We denote here a gain of a new active origin as the event of de novo emergence. Alternatively, it could also correspond to the increase in efficiency of an origin undetectable in other species. Our definition of an origin loss is the inactivation of a previously active origin at the chromosomal level, regardless of its capacity to sustain the autonomous replication of a plasmid (ARS activity). We also cannot rule out that an origin loss would correspond either to the reduction of its chromosomal activity to a level below the sensitivity of the experiment or to the rise of an earlier origin nearby, resulting in passive replication of the considered

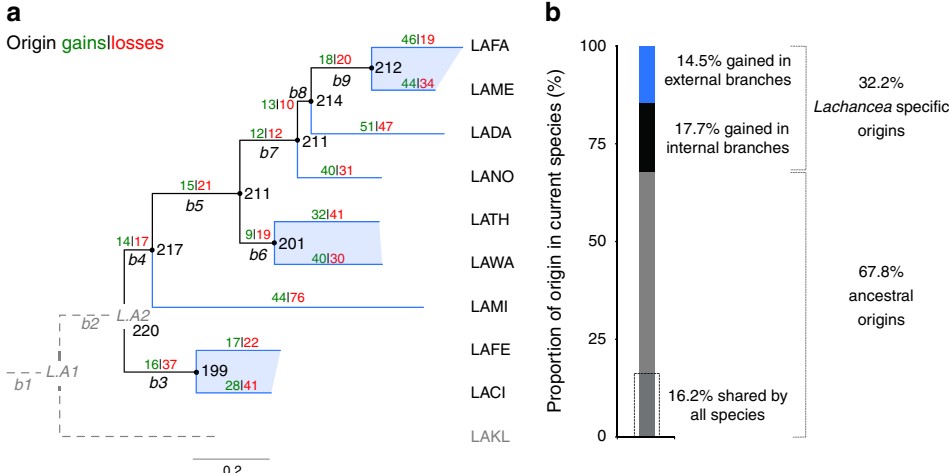

**Fig. 3** Evolution of the repertoire of active replication origins in the genus *Lachancea*. **a** The phylogeny of the ten *Lachancea* species is taken from ref. [31]. Abbreviations of the species names: LAFA: *L. fantastica*, LAME: *L. meyersii*, LADA: *L. dasiensis*, LANO: *L. nothofagi*, LATH: *L. thermotolerans*, LAWA: *L. waltii*, LAMI: *L. mirantina*, LAFE: *L. fermentati*, LACI: *L. cidri* and LAKL: *L. kluyveri*. LAKL was used as the outgroup species; therefore evolutionary events that occurred on both the LAKL and the *b2* branches (dotted lines) could not be retraced. Internal branches, labeled *b3* to *b9*, and terminal branches are drawn in black and blue, respectively. The number of origin gains (in green) and losses (in red) were estimated for each branch of the tree under a birth-death evolutionary model. The inferred number of active replication origins in the ancestral genomes is indicated next to the corresponding node of the tree. **b** The histogram indicates the proportion of active origins that were vertically inherited from the *L.A2* ancestor (in gray), gained on internal branches of the tree (in black) and gained on terminal branches (in blue). The proportion of ancestral origins retained in all nine genomes is indicated by the open frame

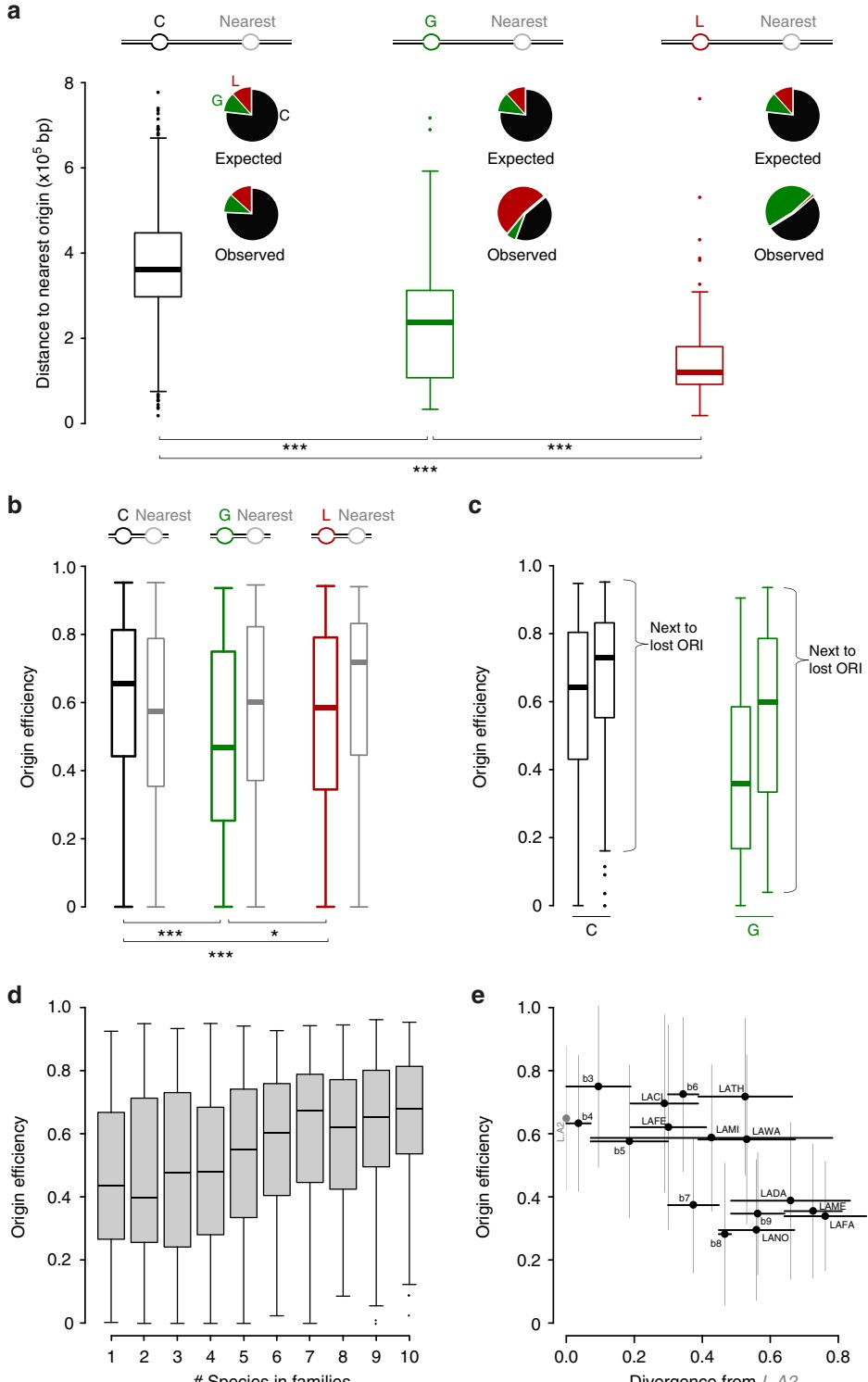

origin. We located on the phylogenetic tree 477 losses and 439 gains since the nine *Lachancea* species diverged from *L.A2* (Fig. 3a). These conservative figures exclude 94 events that occurred on the two most internal branches of the tree (*b2* and *L. kluyveri*), for which it was not possible to discriminate an origin loss on one branch from an origin gain on the other. The 439 gain events resulted in 623 *Lachancea*-specific origins because a single gain event occurring on an internal branch of the tree results in several origins present in different descendant species. These

origins represent 32% of all active replication origins in the nine *Lachancea* genomes today.

The number of origin losses per branch, and to a lesser extent the number of gains, significantly correlate with branch length, suggesting that the accumulation of point mutations could have had an impact on the dynamics of the set of active origins (Supplementary Fig. 10a). Additionally origin losses and gains also correlate with the number of inversions, translocations, and duplications per branch[31] (Supplementary Fig. 10b, c). Although

there is no obvious causal relationship between these observations, they reveal that protein divergence, chromosome architecture, and replication programs evolve in a coordinated manner.

**Origin losses and gains are linked both in time and space**. To further characterize the evolutionary dynamics of active replication origins, we compared the physical and functional properties between conserved, lost, and gained origins. To ensure we used an appropriate proxy of the properties at the time they were gained or lost, we focused on the most recent events that occurred along the *Lachancea* phylogeny comparing sister species within the three most closely related pairs (*L. fantastica*/*L. meyersii*, *L. thermotolerans*/*L. waltii* and *L. fermentati*/*L. cidri*, see blue shaded area in Fig. 3a). After filtering for dubious cases of losses and gains (see Methods), this data set comprises 997 conserved, 151 gained, and 151 lost origins. As physical properties, we studied the distance between each origin and its centromere, closest telomere, closest genome rearrangement breakpoint, and closest neighboring origin. As functional properties, we considered efficiency and characteristic firing time of each individual origin and that of its closest neighboring origin. We derived the efficiency and firing time of each individual origin in the ten *Lachancea* genomes by fitting a stochastic mathematical model to our replication timing data[14] (see Methods). For lost origins, we inferred their characteristics based on the corresponding features of their orthologous origins in the sister genomes.

Firstly, we found no difference in the origin-centromere and origin-telomere distances between the conserved, gained, and lost origins, showing that their relative chromosomal location did not influence their evolutionary fate (Supplementary Fig. 11a). Secondly, using 74 rearrangements that occurred along the six terminal branches[31] we found no clear association with replication origins, neither in the first 5 kb that directly flank the breakpoints nor further away (Supplementary Fig. 11b). Similarly, for each origin category (conserved, gained or lost), we found no association with the breakpoints (Supplementary Fig. 11c). All these results suggest that chromosomal rearrangements play little role, if any, in the evolutionary dynamics of replication origins and confirm our previous findings that replication origins and synteny breakpoints do not significantly collocate in *L. kluyveri* and *L. waltii* genomes[23]. Lastly, we found that origin losses and to a lesser extent gains tend to occur closer to their nearest origin than conserved origins (median distances of 13 kb and 24 kb vs 36 kb, respectively, Fig. 4a), indicating that origins tend to be lost when they are in close vicinity to another replication origin. Moreover, we found a strong physical association between lost and newly gained origins (Fig. 4a). These results show that, within the same phylogenetic branch of the tree, active replication origins are preferentially lost when they are in the close vicinity of a newborn origin.

Next we compared the efficiencies between the three origin categories and found that gained and to a lesser extent lost origins have lower efficiencies than conserved (Fig. 4b). We also found that the efficiencies of the nearest origins directly flanking lost origins are higher than those flanking both gained and conserved. At first this may seem paradoxical because it suggests that an origin would be preferentially lost when it is flanked by a gained and highly efficient origin, while gained origins have in general the lowest efficiencies of all three categories (Fig. 4b). However, we found that the subset of both gained and conserved origins in close vicinity to lost origins include the most efficient origins within their respective category (Fig. 4c).

Therefore, a clear picture of the evolutionary dynamics of replication origins emerges from these analyses, where origins are preferentially lost when located close to a newborn origin emerging with a high firing efficiency. Similar results were obtained with characteristic firing times, as late firing origins are preferentially lost near early firing origins that have emerged recently on the same phylogenetic branch (Supplementary Fig. 12).

**Mechanisms of origin losses**. To get information on the relative rates at which conserved and gained origins would be lost, we looked at their type (conserved or gained) in ancestral branches *b3*, *b6*, and *b9* (Fig. 3a), i.e. immediately prior to their loss in terminal branches. For the 79 cases of losses that occurred next to a conserved origin, we found that the origin lost in the terminal branches was systematically a conserved origin in the ancestral species, suggesting that old origins are more likely to be lost than new origins ($P = 0.05$, Supplementary Fig. 13).

To further characterize the mechanisms by which origins are lost, we compared our set of 30 origin losses in the *L. waltii* branch to an independent dataset previously generated in this species[3]. The authors identified 194 ARS, of which 156 were shown to function as chromosomal replication origins during S-phase[3]. We found that 21 of our losses did not overlap with any ARS and 4 corresponded to an ARS but were devoid of chromosomal activity. The five remaining cases of origin losses in our study matched chromosomally active origins in the previous dataset (17%, by comparison 79% of the 156 active origins were also detected in our experiments). Therefore, a minority of our loss events would correspond to low efficiency origins that escaped detection because the proximity of a highly active origin. The remaining 83% correspond to true origin losses that could occur through the loss of ARS activity. In order to experimentally confirm this result, we performed an ARS assay for 16 loci of the *L. thermotolerans* genome corresponding to

**Fig. 4** Physical and functional properties of lost, gained, and conserved replication origins. The original dataset comprises 1073 conserved, 207 gained, and 187 lost origins; however, after filtering for dubious cases the number of total gain and loss is both 151 events. In all panels, *** and * represent $P < 10^{-03}$ and $P < 5 \times 10^{-02}$, respectively, using a chi-square two-sample test. **a** Distribution of the distance separating Conserved (C), Gain (G), and Lost (L) origins from their nearest origins. The pie charts show the expected, i.e. the theoretical percentages if the nearest origins were randomly sampled in the population of origins and the observed proportions of C, G, and L nearest origins for the three categories. **b** Distribution of the efficiencies of C, G, and L origins and of their nearest origins. **c** Split distributions of the efficiencies of C and G origins based on the category of their nearest origin. **d** Distribution of the firing efficiencies as a function of the number of *Lachancea* species comprised into the families of orthologous origins ($R^2 = 0.891$ on median values, $P = 4.1 \times 10^{-05}$). For boxplots in **a–d**, whiskers are defined by selecting the values within the range of the 75th and 25th percentile plus or minus 1.5 x (75th percentile – 25th percentile) respectively, and at the maximum distance from the median. **e** Correlation between origin efficiency and origin age for 546 origin families comprising 220 ancestral and 326 *Lachancea*-specific families ($R^2 = 0.44$, $P = 1.3 \times 10^{-03}$). Each dot represents the median efficiency of all the replication origins that appeared in a given branch of the phylogenetic tree. The x-axis represents the total branch length between *L.A2* (the last common ancestor of the nine species, indicated in light gray) and the branch of appearance of the new origins, placing the most ancestral origins on the left and the youngest on the right of the plot. Vertical error bars represent the standard deviation of origin efficiencies and horizontal bars represent the span of the branch lengths based on the phylogenetic tree in Fig. 3

experimentally defined active ARS in the *L. waltii* genome[3]. Four of them are defined here as conserved between the two genomes while the remaining 12 correspond to origin losses in *L. thermotolerans*. We found that all four conserved origins show a clear ARS activity (Supplementary Fig. 14). Interestingly, the conserved origin of *L. thermotolerans* with the lowest ARS activity is also the least efficient and latest firing origin. By contrast, only 4 of the 12 loci corresponding to lost origins show ARS activity (Supplementary Fig. 14). The remaining eight loci have lost their ARS activity. We then tested whether the loss of ARS activity could be due to mutations affecting the ACS or its B1 element[2,5,39]. We identified 75 syntenic origins between *L. thermotolerans* and *L. waltii* origins with known ACS[3]. Using MEME[40], we searched for the *L. thermotolerans* ACS and identified a motif of 36 nucleotides highly similar to the ACS and B1 element of both *L. waltii* and *S. cerevisiae*; therefore, we defined it as the *L. thermotolerans* ACS (Supplementary Fig. 15). Using the same approach, we searched for the ACS across the other species, we found a motif highly similar to the *L. waltii/L. thermotolerans* ACS in *L. fantastica* and *L. nothofagi*, a more degenerate motif in *L. meyersii* and *L. dasiensis* and we failed to detect any motif in the four more distantly related species (Supplementary Fig. 15). Interestingly, the ACS of *L. thermotolerans* was identified in the four conserved origins tested for ARS activity. However, this ACS was found in only 6 of the 12 tested lost origins, 3 of the 6 displaying ARS activity. Moreover, the logo resulting from the four conserved origins was highly similar to the ACS while the one derived from the six lost origins was slightly less conserved. For only the three lost origins without any ARS activity, the ACS motif is unrecognizable (Supplementary Fig. 16). These results suggest that the loss of the ARS activity is related to the degeneration of the ACS.

**Evolutionary age impacts active replication origin features.** We found that origin efficiencies positively correlate with the number of species represented in the origin families (Fig. 4d), suggesting a relationship with the evolutionary age of the origins. All individual lineages contributed to this signal, as we observed that the efficiency of species-specific origins is significantly lower than that of ubiquitous origins (Supplementary Fig. 17a). Species-specific origins also fire later than conserved origins (Supplementary Fig. 17b). The relationship between efficiency and origin family size did not result from a lack of precision in origin location for weak and isolated origins as compared to strong and ubiquitous origins (Supplementary Fig. 17c, d). The number of species per family might not be necessarily the best proxy for the evolutionary age of the origins as a family comprising four species could be older than a family that comprises five if the former underwent more origin losses than the latter. In addition, species-specific origins could in fact have different ages because of the variation in length between the terminal branches of the tree. In order to remove these confounding factors, we used our reconstruction of replication origin history (Fig. 3a) to assign a phylogenetic age to each origin family, corresponding to the cumulated branch lengths between *L.A2* and its branch of origination in the phylogenetic tree. We found a correlation between the phylogenetic age and the efficiency of the origin families (Fig. 4e). Both the ancestral and oldest *Lachancea*-specific origins, originating from *L.A2* and the *b3/b4* branches respectively, fire more efficiently on average than the youngest origins that were gained on the small terminal branches leading to *L fantastica* and *L. meyersii* (Fig. 4e). The same was observed with characteristic firing times (Supplementary Fig. 18). Therefore these results show that origin activity correlates with the evolutionary age of the replication origins.

## Discussion

Our study is the first detailed reconstruction of the evolutionary history of genome replication in eukaryotes. Using the model yeast genus *Lachancea*, which exhibit a continuous evolutionary range from closely related to more divergent genomes, we captured all intermediate states between highly conserved and significantly reprogrammed temporal orders of genome replication. The replication-timing program evolves coordinately with protein sequence and chromosome architecture (Fig. 1c and Supplementary Fig. 10). Previous data suggested that replication origins tend to colocalize with synteny breakpoints between yeast species more distantly related than *Lachancea* species[23,28,41,42]. However, we surprisingly discovered that the accumulation of chromosomal rearrangements did not drive the evolution of the replication

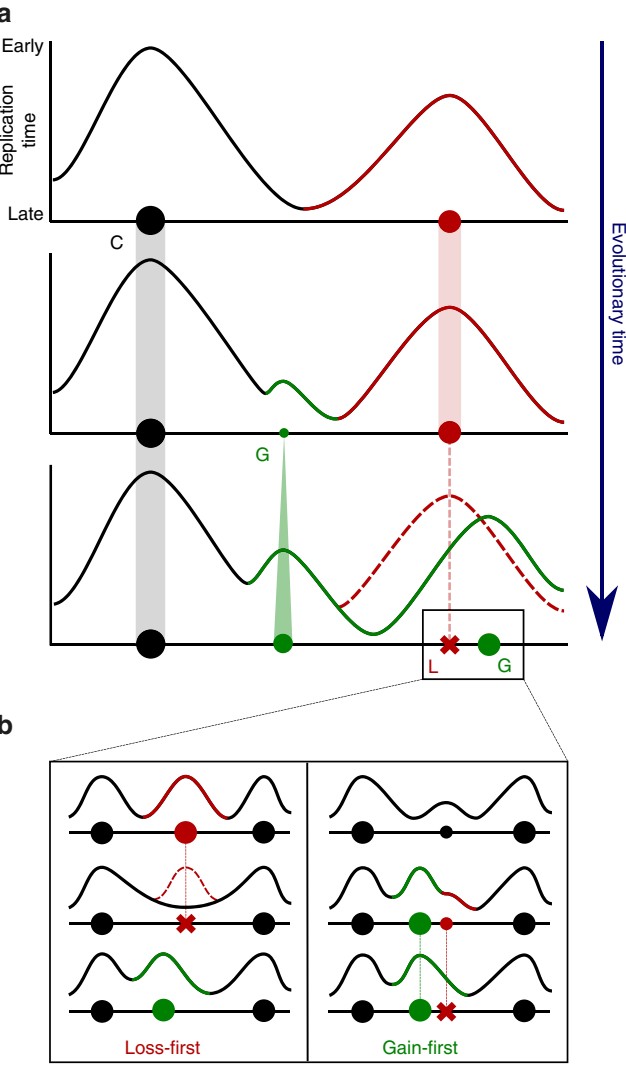

**Fig. 5** Evolutionary model of the temporal program of genome replication. **a** The diagram illustrates five principles governing the dynamics of replication origins at large evolutionary scale: (i) chromosomally active replication origins are continuously gained (G) and lost (L) during evolution, (ii) conserved origins (C) are more ancestral and have on average stronger activity than younger origins, (iii) newly gained origins have on average low activity and their strength increases over evolutionary times, (iv) conserved origins are preferentially lost and (v) origin loss and gain of strong origins occur in a direct chromosomal vicinity. **b** The two pathways called "Loss-first" and "Gain-first" describe, at shorter evolutionary distances, the coordinated gain of a strong origin with the loss of a strong (left) or weak (right) origin

program (Fig. 2b and Supplementary Fig. 11b, c). Here, we found that the dominant process in replication profile evolution is the highly dynamic evolutionary renewal of the cohort of replication origins across species (Fig. 2d). The evolutionary dynamics of replication origins in *Lachancea* far exceeds that of chromosome rearrangements with 394 recent events of origin gains and losses compared to only 74 translocations and inversions that reached fixation on the terminal branches of the tree[31] (Fig. 3a). The small proportion, 16.2%, of ancestral origins belonging to families that remained conserved across all *Lachancea* genomes also illustrates the high evolutionary turnover of active replication origins. These results are in marked contrast with the high conservation in the location and activation time of chromosomally active origins between much more closely related species from the *Saccharomyces* sensu stricto complex[22]. In total, we characterized 1010 gain and loss events since the *Lachancea* species diverged from their last common ancestor. The chromosomally active replication origins detected by our genome-wide timing survey only correspond to a subset of a pool of possible ARS because cells, in order to overcome potentially irreversible double fork stalling events, license many more origins than they use[1–3,43–45]. For instance, in *S. cerevisiae*, the experimental deletion of all efficient replication origins from entire chromosomes only causes a marginal mitotic instability because dormant origins become active and contribute to the replication of the genome[46,47]. The evolutionary dynamics of replication origins uncovered here is not directly comparable to this ARS modularity because time scales are different.

However, we cannot rule out that the evolutionary gains of new chromosomally active origins could in fact correspond to the activation of dormant ARS or to the increase in efficiency of a weak origin. Concerning origin losses, we identified two categories differing by their ARS activity. We observed that the origin losses that concomitantly lost their ARS activity were associated with an important degeneration of the ACS while origin losses that retained ARS activity show greater ACS conservation (Supplementary Fig. 16). The origin loss, in the former case, could result from mutations acting in *cis* in the ACS or its flanking B1 element that would impair their recognition by ORC thereby abolishing the ARS activity[2]. In the latter case, it could result from mutations acting in *trans* and affecting neighboring gene expression and/or chromatin states and thereby altering ORC binding in the chromosomal context only. The key result of our study is that we uncovered several principles that governed the gain, loss and conservation of replication origins during genome evolution (Fig. 5a). Firstly, new chromosomally active replication origins are continuously gained and lost during genome evolution (Fig. 3 and Supplementary Figs. 9 and 10). Secondly, the activity of replication origins depends on their evolutionary age. New origins gained on terminal branches of the tree emerge with globally low efficiencies and late firing times (Fig. 4b and Supplementary Fig. 12a), conserved origins that appeared on internal branches have intermediate activity and ancestral origins are the strongest (Fig. 4e and Supplementary Fig. 18b). These findings suggest that origin activity becomes stronger while they age over long evolutionary periods. Alternatively, natural selection would rapidly purge the genomes of low efficiency replication origins and select for the few new origins that emerged with high efficiency. However, this hypothesis seems less likely because, even though lost origins have on average intermediate activity relative to conserved and gained origins, they also cover the entire range of efficiency (Fig. 4b). Thirdly, we found that origins lost in the terminal branches of the tree systematically corresponded to conserved origins, suggesting that old origins are more likely to be lost than new origins. Finally, the subset of gained origins corresponding to the most efficient and earliest firing are physically associated with lost origins, both

in terms of physical distance along the chromosome and time of appearance, i.e. in the same branch of the phylogenetic tree (Fig. 4a−c and Supplementary Fig. 12b). Similarly, conserved origins that flank lost origins are also on average the most efficient of all conserved origins (Fig. 4b, c).

Altogether, the above observations fit within a two-pathway model of origin gain and loss coordination during evolution that we call Loss-first and Gain-first. In the Loss-first pathway, the main cause explaining the gain of a strong active origin would be the initial loss, nearby along the chromosome, of a strong origin, generating a large region devoid of initiation zone (Fig. 5b). Conversely, in the Gain-first pathway, the main cause explaining the evolutionary loss of a weaker replication origin would be the appearance nearby of a stronger replication origin (Fig. 5b). In a nutshell, our model relies on five simple principles (Fig. 5): (i) chromosomally active replication origins are continuously gained and lost during evolution, (ii) conserved origins have on average stronger activity than younger origins, (iii) newly gained origins have on average low activity and their strength increases over evolutionary time, (iv) conserved origins are preferentially lost and (v) origin loss and gain of strong origins occur in a direct chromosomal vicinity. It is intriguing to speculate on the physiological principles and constraints responsible for the evolutionary tradeoffs leading to these principles. Interestingly, despite their high evolutionary turnover, chromosomally active replication origins remain more regularly spaced than expected in all ten *Lachancea* genomes since they diverged from their last common ancestor approximately 80 million years ago (Supplementary Fig. 3d). Regular spacing of replication origins along chromosomes was also reported for other yeast species[36,48–50]. These findings suggest that origin distribution has been optimized to limit large inter-origin distances where irreversible double fork stall events are more likely[36]. If natural selection acts to keep active replication origins regularly spaced along chromosomes, it means that there may be a cost for both low and high origin density. Alternatively, since there is some rate of loss of origins, simply having no benefit for high density would ensure that over time origins in high-density regions are lost.

In addition, even if chromosomal rearrangements play no direct role in the evolutionary dynamics of origins, they can change the distance between origins, thereby creating in some instances larger origin-free regions. Therefore origins would need to be able to evolve at least as quickly as chromosomes rearrange. However, in the absence of any chromosomal rearrangements, the molecular determinants that drive the evolutionary loss and gain of active replication origins and therefore the remodeling of the temporal program of genome replication remain so far unidentified.

## Methods

**Yeast strains and growth conditions**. All yeast strains and growth conditions are summarized in Supplementary Table 1. For all *Lachancea* species, excluding *L. nothofagi* and *L. fantastica*, cells were grown at 30 °C in YPD broth (BD-Difco). *L. nothofagi* and *L. fantastica* were grown at 24 °C since, at 30 °C, the former does not grow and the latter aggregates. All time-course experiments were performed at 23 °C in YPD.

**DNA sample preparation**. The experimental and analytical methods used for the time course experiments are fully described in ref. [34]. Briefly, G1 cells were isolated from an asynchronous cell culture using centrifugal elutriation, and then grown at 23 °C in YPD. Time-point samples were taken regularly until the cells reached the G2 phase. Progression of the cells through the cell cycle was monitored using flow cytometry, as described in ref. [23]. Samples covering the whole S-phase were selected and DNA was extracted using the genomic-tip 20/G isolation kit (Qiagen). For the MFA experiments, cells were grown in YPD at the appropriate temperature. The exponential (Expo) and stationary phase (Stat) samples were collected after 4 and 30 h of growth, respectively. DNA was extracted using the genomic-tip 20/G isolation kit (Qiagen).

**Deep sequencing**. For each sample, a minimum of 300 ng of genomic DNA was sequenced as 50 bp single reads using Illumina technology (Single-End, 50 bp). A minimum of 10 million and 15 million reads per sample were used for the time-course experiment and the MFA experiment, respectively. To avoid differential PCR biases between samples in the time-course experiment, all multiplexed libraries were pooled before PCR amplification as described in ref. [23]. Libraries were de-multiplexed and adaptator sequences were removed from the reads. Sequences were remapped to the reference genomes[31–33] using BWA (0.59) and allowing no mismatch and no gap. Mapped reads were subsequently filtered to keep only unique match and high-quality mapping scores (MAPQ > 37, i.e. base call accuracy >99.98%).

**Mean replication time and MFA profiles**. Mean replication times were calculated from time-course experiment sequences. Reads were counted in 500 bp non-overlapping windows, along the genome. Changes in DNA copy number were measured by calculating the ratio of the number of sequences between S-phase samples and the reference sample, corresponding to G1 or G2 phase. For each time point, the median of the S/G1 or S/G2 ratio was adjusted to correspond to the DNA content measured by flow cytometry during S-phase progression. Subsequently, ratios were re-scaled between 1 and 2 and for each window, the time where the scaled ratio equaled 1.5 was defined as the Trep. Finally, mean replication times were obtained by smoothing the data with a loess regression (see ref. [34] for details).

For the MFA experiments, reads were counted in all 500 bp non-overlapping windows, along the genome. Windows where the number of sequences was defined as an outlier (> or <1.5 times interquartile spaces) were filtered out. The MFA ratio is calculated by dividing, for each window, the number of sequences from the Expo sample by the number of sequences from the Stat sample as described in ref. [35]. The MFA profile is obtained by smoothing the data with a loess regression. All timing data are available in the Supplementary Data 1.

**Identification of replication origins**. The mean replication times and MFA profiles were plotted as a function of the chromosomal coordinates. The first derivative of these curves was estimated by calculating the slope of each coordinate $x$ of the window $i$ along the genome, using the following formula $(y_i - y_{i-1})/(x_i - x_{i-1})$, as described in ref. [34]. The second derivative was then estimated from the first derivative values using the same method and plotted as a function of the chromosomal coordinates. Both peaks and shoulders from the original mean replication time and MFA curves appear as clear peaks on the second derivative plot. The chromosomal location of these peaks was determined where the slope curves corresponding to the third derivatives intersect 0 from negative to positive value as described in ref. [34]. We defined peak coordinates as the location of active replication origins only when peaks were co-detected in both the mean replication time and the MFA second derivatives profiles. Then, for each replication origin location, we compared its two chromosomal coordinates issued from the two curves to estimate the precision of our experiments in calling origin locations. We found a median and a mean difference between the two peak coordinates of 4.0 and 4.9 kb, respectively. All origin locations are available in the Supplementary Data 1.

**Modelization of replication kinetics**. We employed a 1D nucleation-growth mathematical model of stochastic replication kinetics similar to models available in the literature[12,13] and detailed in ref. [14], using constant probability of origin activation ($\gamma = 0$). Empirical parameters (origin firing rates and replication fork speed) were inferred through fitting experimental data on DNA copy number as a function of position and time with the model (Supplementary Fig. 19). The initial firing rates were obtained from the slope of the first time points in the replication profiles. The fits are performed using the full replication time-course data, by minimizing the distance between the replication timing profiles in the model and in the experimental data, by an iterative updating method with adaptive steps[14]. The objective function was defined as the average of squared differences of the experimental and theoretical replication profile. We used the positions of replication origins as defined in the section above and kept them fixed in the fits. The model fit allows the direct estimation of the origin firing rates. Characteristic firing times correspond to the inverse of rates and therefore are independent from the interference from nearby origins. Once the model parameters (origin rates, replication fork speed) are fixed, the computational model was run to simulate S phases of single cells. For each origin, we determined in how many simulation runs it was actively or passively replicated. The fraction of simulation runs where the origin was actively replicated is our estimation of the origin efficiency. This procedure is identical to the one defined by Nieduszynski and coworkers[13].

**Comparison of temporal programs of genome replication**. The identification of orthologous genes and the conservation of synteny blocks were computed with the SynChro algorithm[51] for all pairwise combinations between the ten *Lachancea* species, the four *Saccharomyces* and the *C. albicans* species. Genome annotations were downloaded from GRYC (http://gryc.inra.fr), saccharomycessensustricto (http://www.saccharomycessensustricto.org), and CGD (http://www.candidagenome.org). We used our mean replication timing profiles for the *Lachancea* species and the previously published replication data for the *Saccharomyces* species and *C. albicans*[22,25]. For each pairwise combination, a Spearman's

correlation coefficient (rho) is calculated on replication timing for all orthologous pairs. As a null model, the same correlation was calculated after applying an offset on orthologous coordinates for one of the two species of the comparison. The offset was re-defined 100 times and correlations were calculated for each combination of an offset profile from one species and an original profile from the other species.

**Construction of replication origin families**. We used synteny conservation between the two protein coding genes that flank each origin position to construct families of orthologous active origins. Because of the poor synteny conservation in subtelomeres, we excluded the 96 subtelomeric origins that were interstitially located between the telomeres and the first synteny blocks comprising at least five syntenic genes. We constructed origin families for the remaining 2168 internal origins, representing 96% of the total number of origins (Supplementary Fig. 7a). For all pairwise comparisons between two *Lachancea* species, we first projected the position of each replication origin of one genome onto the chromosomal coordinates of a second genome based on the synteny conservation of its two flanking coding genes, and reciprocally from the second genome to the first one. Each projection was associated with its nearest resident origin and then, two origins were defined as conserved between two species when the projected and the resident origins were located at most two syntenic genes apart, in both directions (delta = 2, Supplementary Fig. 7b). The resulting 3730 pairs of conserved orthologous origins were subsequently clustered into origin families which simply correspond to the assembly of all pairs of orthologous origins into connected components, such as those illustrated in Supplementary Fig. 7c. Active replication origins distributed into 374 multi-species origin families comprising 1956 origins (90%) and 212 species-specific singleton origins (10%, Supplementary Fig. 7a).

To assess the quality of our origin families, we generated a null model where the positions of the origins were randomized, according to the following constraints:

- randomized inter-origin distances must preserve the same distribution than the observed inter-origin distances. This property is particularly important, since actual origins are more regularly spaced than randomly distributed origins (Supplementary Fig. 3d);
- the position of the first origin on each chromosome must be located between the beginning of the chromosome and two-times the position of the first actual origin;
- the relative proportions of intra- and inter-genic origins must be conserved by the randomization procedure.

In particular, the first two constraints guarantee that the average number of origins is globally conserved, even if it may slightly fluctuate between samples drawn randomly from the null model.

Thereafter, for each pairwise comparison, origin positions were randomized 100 times in the first genome and projected on the second genome, and reciprocally. Randomized origin families were then constructed with the same procedure as for the real origin families. We used the randomized families to define the optimal number of syntenic genes allowed between two conserved origins. The threshold of Delta = 2 syntenic genes was determined after testing all values between 0 and 6 intervening genes and looking for the value that maximized the differences between real origin families and null model. We found that Delta = 2 was the value that jointly (i) maximized the number of conserved origins in the real versus the random dataset, (ii) maximized the number of families comprising ten origins (one per species) in real vs random dataset comparisons and (iii) minimized the number of families with more than ten members (Supplementary Fig. 7d). Finally, we also found that Delta = 2 limited the proportion of families with more than one origin per species (1.9% for Delta = 2 vs 8% for Delta = 3).

To check that our methodology indeed captured a true evolutionary signal corresponding to the orthology relationship between replication origins, we compared the distributions of the number of origins per family between the real and the random dataset and found that they were clearly different from what is expected from a null model (Supplementary Fig. 8a). Moreover, we constructed two phylogenetic trees based on the composition of the real origin families in the ten *Lachancea* species. First, we performed hierarchical clustering implemented in MEV (http://www.tm4.org./#/welcome) based on the phylostratigraphic patterns of origin families. Second, we built a distance matrix representing the proportion of conserved origins between any two pairs of species and generated an NJ tree, using Phylip version 3.695[52]. The two resulting tree topologies are very similar to the topology of the reference species tree based on the concatenation of 3598 orthologous protein sequences[31], with only few bipartitions being different (Supplementary Fig. 8b).

**Inference of replication origin history**. Based on the composition of the origin families, the evolutionary history of origin conservation, gain and loss was inferred using the Gloome online program[53]. As input, we used simplified phyletic patterns limited to the presence (1) or absence (0) of an origin family within *Lachancea* species. We used an evolutionary model where the probability of gain and loss is equal across all sites. We tolerated more than one possible creation event per family. The parsimony cost of the gains was set to 2. Other parameters were set to default values.

To compare the physical and functional properties between conserved, lost, and gained replication origins, we only focused on the three most closely related pairs

of species (*L. fantastica*/*L. meyersii*, *L. thermotolerans*/*L. waltii*, and *L. fermentati*/*L. cidri*). This resulted in an initial dataset of 1280 cases (886 conserved origins, 207, gains and 187 losses). We then filtered from this dataset the dubious cases of losses and gains that correspond to origins that were detected by only one of the two methods (MFA or mean replication time). This resulted in a final dataset composed of 1148 cases (846 conserved origins, 151 gains, and 151 losses).

**ARS assay.** Using the Gibson assembly method[54], we constructed 16 plasmids containing a conserved (4 cases) or lost origin (12 cases). These 16 loci were selected because they showed strict synteny conservation with previously published chromosomally active *L. waltii* ARS[3]. For each ARS, the exact location of the ACS in *L. waltii* was used to identify the corresponding syntenic intergene in *L. thermotolerans*. These intergenes were amplified by PCR using the oligonucleotides described in Supplementary Table 2. We used as a backbone the pRS41K plasmid where the native *S. cerevisiae* centromere was replaced by the centromere of *L. thermotolerans* chromosome 0C. The *S. cerevisiae* replication origin originally present in pRS41K was substituted in each plasmid by one of the 16 *L. thermotolerans* PCR fragments. Plasmids were transformed into *L. thermotolerans* cells using the LiAc/PEG method[55]. Cells were plated on YPD with G418 (200 μg/mL) and growth for 4 days at 30 °C and then the number of colonies and their size were measured.

**ACS detection.** We selected 123 chromosomally active origins in *L. waltii* that corresponded to known ARS that were previously published[3]. We then selected in the genomes of all other *Lachancea* species, the replication origins that were conserved in families with these 123 origins. We used the known ACS position in *L. waltii* to identify the corresponding gene/intergene strictly conserved in synteny in the other species. Based on these datasets, we use MEME (zoops, size 9-40, e-value threshold 10) to search for motives, both in *L. waltii*, as a control, and in the other *Lachancea* species. The number of occurrences containing the detected motif was then evaluated using MAST (e-value threshold $10^{-4}$)[40].

**Data availability.** The fastq files are deposited in the Sequence Read Archive with the project number SRP111158 under accessions ranging from SRR5807795 to SRR5807891.

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

## Acknowledgements

This work was supported by the Agence Nationale de la Recherche (GB-3G, ANR-10-BLAN-1606 and Phenovar, ANR-16-CE12-0019). We thank our colleagues Gianni Liti, Bertrand Llorente, Carolin Müller, Conrad Nieduszynski, and Samuel O'Donnell for fruitful discussions and constructive suggestions.

## Author contributions

G.F., N.A., and S.D. wrote the paper. M.C.-L. and M.W. contributed to the manuscript. G.F. supervised the project. G.F. and N.A. designed the experiments. N.A. and S.D. performed the experiments. N.A. performed the analyses. M.W. and A.F. contributed to the analyses. Q.Z. and M.C.-L. performed the mathematical modeling. Y.J., E.v.D. and C.T. performed the sequencing.

## Additional information

**Competing interests:** The authors declare no competing interests.

