## [Peer Review File · Nature Communications]

Reviewers' comments:

Reviewer #1 (Remarks to the Author):

Agier et al. present the analysis of replication profiling data across the genus *Lachancea*. The phylogenetic diversity of this clade allows them to compare replication profiles, and deduced location and efficiency of the origins that underlie these profiles, between species that range from having very similar to quite distinct profiles. These comparisons elucidate the evolution of origin function over 80 million years. Their analysis demonstrates that origins are dynamic and plastic genetic elements that rapidly change function with little apparent evolutionary constraint. The major selection force seems to be the requirement to avoid large regions devoid of origin function. These conclusions, and the data that support them, fill an important gap in our knowledge of origin function and evolution and, as such, will be of significant interest to a wide range of readers interested in DNA replication, genome stability and evolution. Nonetheless, the manuscript would be improved by attention to the following points.

The authors refer to origins that they cannot detect as dormant origins. This is unfortunate nomenclature. "Dormant origin" can imply a qualitatively different class of origin that never fires in normal S phase, but is activated to fire by replication stress. What the authors mean here is an origin with an efficiency below their threshold of detection. It would be clearer if they simply referred to them as "inefficient origins". They actually use both nomenclatures in one paragraph.

"Note that what we call here a gain of a new active origin could correspond to the activation of a dormant origin. Similarly, what we call an origin loss could correspond to the inactivation of previously active origin or to the reduction of its activity to a level below the sensitivity of the experiment."

They should rephrase the former sentence as follows.

"Note that what we call here a gain of a new active origin could correspond to the increase in efficiency of a previously undetectable origin."

It was unclear to me if there are many instances in which a new origin and its old neighbor coexist in an ancestral species (b3-b9) before one is lost in an extant species. If so, it would be interesting to know if the new or old origin is more likely to be lost. The simple model predicts the loss of either would be equally likely. If either one is more likely to be lost, it would say something interesting about the selection on old versus new origins. If such pairs are not often found, it would say something interesting about the rate at which origin pairs are lost.

The authors point out the loss of an origin in their analysis could be due to the increase in efficiency of a neighboring origin masking its function. They argue that this is not often the case, but I am not persuaded by their argument. They point out that only 5 of 30 of the origins lost in *L. waltii* colocalized with independently identified ARSs found to be active in the *L. waltii* genome (and therefore apparently masked by an efficient neighboring origin instead of inactivated by mutation). However, they excluded from their analysis 38 ARSs that are not detectably active in the *L. waltii* genome. It is plausible that this group contains origins that are masked by an efficient neighboring origin and thus appear inactive in the chromosome. This class of ARSs should be included in their analysis.

The authors conclude that "there are costs for keeping origins too close, as well as for keeping a higher and lower density of active origins." The cost of low density is clear, but I am not convinced that high density would need to have a cost. Since there is some rate of loss of origins, simply having no benefit for high density would ensure that over time origins in high-density regions are lost.

The authors state that "chromosomal rearrangements play little role, if any, in the evolutionary dynamics of replication origins". Although correct in the context of the statement, it might be worth pointing out, perhaps in the Discussion, that chromosomal rearrangements increase the distance between origins, creating larger origin-free region, and may explain why origins need to be able to evolve at least as quickly as chromosomes rearrange.

Reviewer #2 (Remarks to the Author):

The paper presents the first reconstruction of the evolutionary history of eukaryotic replication origins, by analysing the distributions of replication origins in the yeast genus *Lachancea*. It shows that over evolutionary time, replication origins appear and disappear in the genome independently of major chromosome rearrangements. Origin dynamics over evolutionary time appear to maintain chromosomes in a state where origins are relatively evenly distributed along the chromosomes, which is predicted to reduce the probability of lethal double fork stall events. This work provides important new insights into how natural selection can maintain genome stability in the face of dynamically changing chromosomes. The conclusions are very clear and are presented in an easily understood way. I recommend publication following some relatively small improvements.

It is interesting that timing correlations are weaker around synteny breakpoints than within conserved synteny regions (Fig 2b). I am unclear why the global rho does not lie between the 'synteny break' and 'no synteny break' lines. Does this mean that there are regions of high correlation that are >25 genes away for the synteny break points (the regions used for the definition of 'no synteny')? This is important because a simple explanation for the low correlation of synteny break regions is that when fusions occur between regions with different replication timing, passive replication of the later region by a fork from the earlier region causes the later region to replicate earlier than it otherwise would. Is it possible to model how important this is as a contribution to the lack of correlation at the synteny break points? What happens to the correlation as it is measured at progressively larger distances from the break points?

The authors argue that because the divergence of the timing programme and the loss of conserved origins both decline with phylogenetic distance "... the appearance and disappearance of active replication origins would be the dominant process for shaping replication profiles during evolution." I don't think I agree with this: the timing programme and conservation of origins could both independently decline with phylogenetic distance, without there necessarily being a causal relationship between them.

Minor Points

1. I'd be interested to see the R2 values for individual species in a graph or table in supplementary Fig S3, or added to Table 1.
2. Syntenic homologs and synteny blocks (black and open circles) should be labelled directly in Fig 2a.
3. Page 8: "Similarly, what we call an origin loss could correspond to the inactivation of previously active origin or to the reduction of its activity to a level below the sensitivity of the experiment." It might also be worth pointing out that an origin could appear to be lost if a new origin arises nearby that fires significantly earlier, resulting in passive replication of the old origin (this is the converse of the unmasking of a dormant origin mentioned in the previous sentence).

Reviewer #3 (Remarks to the Author):

Nicolas Agier and colleagues have tackled the question of how the timing program for chromosome replication changes over evolutionary time, using 10 members of the *Lachancea* genus of budding yeasts. This report is the first attempt at systematically characterizing the replication profiles of a large cohort of related species. The authors conducted two independent assessments of replication timing in the 10 species and spend the bulk of the manuscript on the computational analysis of these data. They conclude that there is a high turnover rate for origins characterized by gains and losses that drive changes in the temporal programs of chromosome replication. The work is extremely interesting and novel and could make a significant contribution to our understanding of chromosome evolution.

Concerns with the data and interpretations: In general, we find some of the interpretations to be overstated because supporting data are lacking (see points 1-4 below).

1. Supplementary Figure 2 reveals the overlap in origin designations from the two peak calling methods (MFA and Trep). While the overlap is significant, 20 to 30 percent of peaks found in one method were missing from the other method. They focused only on the common origins to define origin losses and gains. It would be important to know how many, if any, origin losses were actually seen in MFA but not in Trep (or vice-versa). Could many of the proposed origin losses just be a consequence of limitations to their ability to detect low efficiency origins in one or both of the two methods. For example, on page 7 of the results, they comment that they only detected 77% of origins identified in a previous scan of the *L. waltii* genome and that number is almost identical to the overlap (76%) using their two peak calling methods. How many of the called origin loss events are just cases of failed detection in one of the two methods? Include this data in a supplementary figure.

2. While the replication profiles are impressive, the work suffers from a general lack of validation. I would suggest that the authors validate a few cases of origin losses and gains by an alternate method—for example, perform an ARS assay on fragments containing missing origins, do 2-D gels across lost or gained origins in a pair of sister species, and/or compare sequences across the species that have retained an origin and those that have lost the origin to look for sequence-based causes of origin birth or death.

3. The definition of an “origin family” is mysterious. A figure, either a toy figure, or better yet, an example of real data is essential. A figure would also help with the discussion of what constitutes a syntenic origin position. The verbal descriptions (“two origins were defined as conserved between two species when the projected and the resident origins were located at most two syntenic genes apart, in both directions” and “orthologous origins were subsequently clustered into origin families by transitivity”) would benefit from a generic drawing. Our interpretation of their description would seem to indicate that origins that lie in two different, but adjacent intergenic regions are considered conserved. If our interpretation is correct, they appear to be too generous in calling origin conservation—especially if transitivity across rearrangement breakpoints is permitted. Please include a figure of an actually origin family to clear up these issues.

4. The authors make no explicit proposals about what is changing over evolutionary time as origins are lost or gained. There are three possibilities that come to mind: 1) mutations that affect the ARS consensus sequences or its immediate flanking elements (eg. B1, B2 etc) altering their recognition by ORC; 2) mutations that affect neighboring gene expression and/or chromatin states and thereby alter nucleosome occupancy or transcription factor binding in the vicinity of origins and thereby influence ORC binding; 3) mutations that act in trans—such as point mutations to replication initiation proteins (ORC/MCM, etc). Multiple species alignments to detect origin conservation (as performed by Nieduszynski et al, 2006) could shed light on this issue. And one glaring omission is the authors did not attempt to construct ARS consensus sequences across the *Lachancea* lineage. How much divergence has occurred in these 10 species? Are we to assume that they all use the same consensus as *L. waltii* and *L. kluyveri*? This analysis would help distinguish

the three possibilities above.

5. Clarifications needed on figures.

Figure 5: divide into part A and part B. Why are the two graphs (in part A) not aligned when we are looking at the identical region over time?

Supplemental Figure 2: please show the MFA profiles for all chromosomes from all species—potentially including them as overlays in Supplemental Figure 1.

Supplementary Figure 7: The legend indicates that P values are included. They are missing from the figure.

Supplementary Figure 8b: While their simulations of origin locations relative to syntenic breakpoints may not pass a significance cutoff, there does seem to be a trend to find more origins near breakpoints. Perhaps the authors could comment on this trend given other data from the literature.

We are grateful to the reviewers for their positive appraisal of our work, insightful comments and constructive suggestions that we have taken carefully into account, as detailed below.

Reviewers' comments:

Reviewer #1 (Remarks to the Author):

Agier et al. present the analysis of replication profiling data across the genus *Lachancea*. The phylogenetic diversity of this clade allows them to compare replication profiles, and deduced location and efficiency of the origins that underlie these profiles, between species that range from having very similar to quite distinct profiles. These comparisons elucidate the evolution of origin function over 80 million years. Their analysis demonstrates that origins are dynamic and plastic genetic elements that rapidly change function with little apparent evolutionary constraint. The major selection force seems to be the requirement to avoid large regions devoid of origin function. These conclusions, and the data that support them, fill an important gap in our knowledge of origin function and evolution and, as such, will be of significant interest to a wide range of readers interested in DNA replication, genome stability and evolution. Nonetheless, the manuscript would be improved by attention to the following points.

The authors refer to origins that they cannot detect as dormant origins. This is unfortunate nomenclature. "Dormant origin" can imply a qualitatively different class of origin that never fires in normal S phase, but is activated to fire by replication stress. What the authors mean here is an origin with an efficiency below their threshold of detection. It would be clearer if they simply referred to them as "inefficient origins". They actually use both nomenclatures in one paragraph.

"Note that what we call here a gain of a new active origin could correspond to the activation of a dormant origin. Similarly, what we call an origin loss could correspond to the inactivation of previously active origin or to the reduction of its activity to a level below the sensitivity of the experiment."

RESPONSE: We changed all instances of 'dormant origins' into 'inefficient origins' or 'weak origins' when referring to origins that we cannot detect.

They should rephrase the former sentence as follows.

"Note that what we call here a gain of a new active origin could correspond to the increase in efficiency of a previously undetectable origin."

RESPONSE: We changed the text accordingly.

It was unclear to me if there are many instances in which a new origin and its old neighbor coexist in an ancestral species (b3-b9) before one is lost in an extant species. If so, it would be

interesting to know if the new or old origin is more likely to be lost. The simple model predicts the loss of either would be equally likely. If either one is more likely to be lost, it would say something interesting about the selection on old versus new origins. If such pairs are not often found, it would say something interesting about the rate at which origin is pairs are lost.

RESPONSE: We thank the reviewer for this insightful question. We performed the suggested analysis and found that old origins seem to be more likely to be lost than new origins. We added a paragraph describing this finding within the Results section (p10) and a new Supplementary Fig. 11.

The authors point out the loss of an origin in their analysis could be due to the increase in efficiency of a neighboring origin masking its function. They argue that this is not often the case, but I am not persuaded by their argument. They point out that only 5 of 30 of the origins lost in *L. waltii* colocalized with independently identified ARSs found to be active in the *L. waltii* genome (and therefore apparently masked by an efficient neighboring origin instead of inactivated by mutation). However, they excluded from their analysis 38 ARSs that are not detectably active in the *L. waltii* genome. It is plausible that this group contains origins that are masked by an efficient neighboring origin and thus appear inactive in the chromosome. This class of ARSs should be included in their analysis.

RESPONSE: We agree with the reviewer and included the set of 38 ARS devoid of chromosomal activity in our analysis. We found that 4 of our losses did correspond to such ARS while the remaining 21 losses did not correspond to any ARS suggesting that origin losses would mainly occur through the loss of the ARS activity. We added a paragraph describing this analysis on p11. See also the response to point 2 of Reviewer 3 on the ARS assay which brings an experimental validation to this conclusion.

The authors conclude that "there are costs for keeping origins too close, as well as for keeping a higher and lower density of active origins." The cost of low density is clear, but I am not convinced that high density would need to have a cost. Since there is some rate of loss of origins, simply having no benefit for high density would ensure that over time origins in high - density regions are lost.

RESPONSE: We agree with this possibility and rephrased the corresponding paragraph on page 16.

The authors state that "chromosomal rearrangements play little role, if any, in the evolutionary dynamics of replication origins". Although correct in the context of the statement, it might be worth pointing out, perhaps in the Discussion, that chromosomal rearrangements increase the distance between origins, creating larger origin-free region, and may explain why origins need to be able to evolve at least as quickly as chromosomes rearrange.

RESPONSE: We rephrased the last paragraph of the discussion to include this suggestion (p16).

Reviewer #2 (Remarks to the Author):

The paper presents the first reconstruction of the evolutionary history of eukaryotic replication origins, by analysing the distributions of replication origins in the yeast genus *Lachancea*. It shows that over evolutionary time, replication origins appear and disappear in the genome independently of major chromosome rearrangements. Origin dynamics over evolutionary time appear to maintain chromosomes in a state where origins are relatively evenly distributed along the chromosomes, which is predicted to reduce the probability of lethal double fork stall events. This work provides important new insights into how natural selection can maintain genome stability in the face of dynamically changing chromosomes. The conclusions are very clear and are presented in an easily understood way. I recommend publication following some relatively small improvements.

It is interesting that timing correlations are weaker around synteny breakpoints than within conserved synteny regions (Fig 2b). I am unclear why the global rho does not lie between the synteny break and no synteny break lines. Does this mean that there are regions of high correlation that are >25 genes away for the synteny break points (the regions used for the definition of no synteny)? This is important because a simple explanation for the low correlation of synteny break regions is that when fusions occur between regions with different replication timing, passive replication of the later region by a fork from the earlier region causes the later region to replicate earlier than it otherwise would. Is it possible to model how important this is as a contribution to the lack of correlation at the synteny break points? What happens to the correlation as it is measured at progressively larger distances from the break points?

RESPONSE: We found that, the correlations gradually increase at progressively larger distances from breakpoints in the range of 5 to 25 genes. We illustrated this trend in a new Supplementary Fig 5 and added this information in the main text (p6-7), relating it to the possibility that this would result from the juxtaposition of two regions with different replication timing, as suggested by the reviewer. However, this does not explain why the global correlation coefficients do not lie between the synteny break and no synteny break lines. We checked that this does not result from regions of high correlation that would be >>25 genes away for the synteny break points, as seen in the new Supplementary Fig 5. It is likely that the reason why the global rho lies above the two other lines is that the sample size used for the global rho is much larger than for the two other datasets. For each pairwise comparison between two species, the global rho is calculated using all pairs of homologous genes while the synteny and breakpoint coefficients are calculated using a much smaller subset of 5 gene windows corresponding to the number of synteny breaks between the 2 genomes.

The authors argue that because the divergence of the timing programme and the loss of conserved origins both decline with phylogenetic distance the appearance and disappearance of active replication origins would be the dominant process for shaping replication profiles during evolution. I don't think I agree with this: the timing programme and conservation of origins could both independently decline with phylogenetic distance, without there necessarily being a causal relationship between them.

RESPONSE: We agree with the reviewer that a correlation does not necessarily imply a causal relationship. However, in the absence of synteny break, there are only two causes that can explain the evolution of the timing program, one is the reprogramming of the origin activation time and the other one is the relocation of origin positions, given that fork velocity is considered to be constant. We saw no effect of the activation time on the evolution of the timing program (Fig. 2c). Therefore, we interpreted the correlation between the conservation of origin position and the timing profile as being causal.

Minor Points

1. I'd be interested to see the R2 values for individual species in a graph or table in supplementary Fig S3, or added to Table 1.

RESPONSE: We added the requested table in Supplementary Fig. 3c

2. Syntenic homologs and synteny blocks (black and open circles) should be labelled directly in Fig 2a.

RESPONSE: This has been done accordingly.

3. Page 8: Similarly, what we call an origin loss could correspond to the inactivation of previously active origin or to the reduction of its activity to a level below the sensitivity of the experiment. It might also be worth pointing out that an origin could appear to be lost if a new origin arises nearby that fires significantly earlier, resulting in passive replication of the old origin (this is the converse of the unmasking of a dormant origin mentioned in the previous sentence).

RESPONSE: We added this possibility in the corresponding paragraph (p8).

Reviewer #3 (Remarks to the Author):

Nicolas Agier and colleagues have tackled the question of how the timing program for chromosome replication changes over evolutionary time, using 10 members of the *Lachancea* genus of budding yeasts. This report is the first attempt at systematically characterizing the replication profiles of a large cohort of related species. The authors conducted two independent assessments of replication timing in the 10 species and spend the bulk of the manuscript on the computational analysis of these data. They conclude that there is a high turnover rate for origins characterized by gains and losses that drive changes in the temporal programs of chromosome replication. The work is extremely interesting and novel and could make a significant contribution to our understanding of chromosome evolution.

Concerns with the data and interpretations: In general, we find some of the interpretations to be overstated because supporting data are lacking (see points 1-4 below).

1. Supplementary Figure 2 reveals the overlap in origin designations from the two peak calling methods (MFA and Trep). While the overlap is significant, 20 to 30 percent of peaks found in one method were missing from the other method. They focused only on the common origins to define origin losses and gains. It would be important to know how many, if any, origin losses were actually seen in MFA but not in Trep (or vice-versa). Could many of the proposed origin losses just be a consequence of limitations to their ability to detect low efficiency origins in one or both of the two methods. For example, on page 7 of the results, they comment that they only detected 77% of origins identified in a previous scan of the *L. waltii* genome and that number is almost identical to the overlap (76%) using their two peak calling methods. How many of the called origin loss events are just cases of failed detection in one of the two methods? Include this data in a supplementary figure.

RESPONSE: As suggested by the reviewer, we looked for origin losses that would in fact correspond to active replication origins that were only detected by a single method. We found 31 such cases out of a total of 187 losses (16.5%). Based on this finding, we filtered out these 31 false positive cases from the set of origin losses and recalculated all functional properties originally presented in Fig 4 (panels a-c). In addition, we applied the same procedure to origin gains. We looked in the genome of the sister species to find whether some of these initially undetected origins were found by only one of the 2 methods. We identified 43 cases out of the 207 origin gains (20.8%). We also filtered out these cases and recalculated the functional properties of origin gains based on this new dataset. The results show the exact same trends than those originally presented. Therefore, instead of including a supplementary figure, we modified the graphs accordingly in Fig 4 (panels a-c), Supplementary Fig 10 and 12 and mentioned the new numbers of conserved, lost and gained origins in the main text (p9). In addition, we explain the filtering step in the method section (p22 in the paragraph 'Inference of replication origin history').

2. While the replication profiles are impressive, the work suffers from a general lack of validation. I would suggest that the authors validate a few cases of origin losses and gains by an alternate method⁵ for example, perform an ARS assay on fragments containing missing origins, do 2-D gels across lost or gained origins in a pair of sister species, and/or compare sequences across the species that have retained an origin and those that have lost the origin to look for sequence-based causes of origin birth or death.

RESPONSE: We thank the reviewer for the positive comments on the profiles and agree that an independent experimental validation would strongly support our findings. Therefore, we performed an ARS assay for 16 regions of the *L. thermotolerans* genome that correspond to experimentally defined ARS in the *L. waltii* genome (DiRienzi et al., 2012). Four of them correspond to conserved origins between the 2 genomes and the remaining 12 correspond to origin losses in *L. thermotolerans*. We first constructed a plasmid containing the centromere of the chromosome OC from *L. thermotolerans* (KLTHOC) using the pRS41k backbone. Then we

cloned the 16 regions in this plasmid and transformed *L. thermotolerans* to test whether these regions were able to sustain replication of the plasmid (we added a paragraph in the Method section (p22)). We found that the 4 regions corresponding to conserved origins show a clear ARS activity (although one has a weaker activity than the others). By contrast, 8 out of the 12 regions corresponding to lost origins show very weak activity (67%). The remaining 4 regions show ARS activity comparable to that of conserved origins. We now describe this experiment in the main text (p11) and added a Supplementary Fig 13 to illustrate the ARS activities.

3. The definition of an origin family is mysterious. A figure, either a toy figure, or better yet, an example of real data is essential. A figure would also help with the discussion of what constitutes a syntenic origin position. The verbal descriptions (two origins were defined as conserved between two species when the projected and the resident origins were located at most two syntenic genes apart, in both directions and orthologous origins were subsequently clustered into origin families by transitivity) would benefit from a generic drawing. Our interpretation of their description would seem to indicate that origins that lie in two different, but adjacent intergenic regions are considered conserved. If our interpretation is correct, they appear to be too generous in calling origin conservation especially if transitivity across rearrangement breakpoints is permitted. Please include a figure of an actual origin family to clear up these issues.

RESPONSE: We apologize for the lack of clarity in our definition. We added two new panels in Supplementary Fig. 7 (b,c) that give a schematic example and provide an illustration of several origin families based on real data. The interpretation of the reviewer is correct, two origins that would map in two different, but adjacent intergenic regions are considered conserved. However, this is not 'too generous' in calling conservation for several reasons. First, our precision in origin location (median of 4 kb, see p19) does not allow us to attribute an origin to a given intergenic region. Secondly, we defined a null model for origin distribution along chromosomes and compared the properties of conserved origin families built with different delta values with that of the null model. We found that this value of $\delta = 2$ which allows the clustering of two origins lying in two neighboring intergenic regions into the same family was clearly the best choice to capture the true evolutionary signal, as thoroughly described in the method section (p20-21). This is a very important aspect of our analysis and we truly believe that we based our decision on objective criteria, providing a robust and optimal clustering method.

4. The authors make no explicit proposals about what is changing over evolutionary time as origins are lost or gained. There are three possibilities that come to mind: 1) mutations that affect the ARS consensus sequences or its immediate flanking elements (eg. B1, B2 etc) altering their recognition by ORC; 2) mutations that affect neighboring gene expression and/or chromatin states and thereby alter nucleosome occupancy or transcription factor binding in the vicinity of origins and thereby influence ORC binding; 3) mutations that act in trans such as point mutations to replication initiation proteins (ORC/MCM, etc). Multiple species alignments to detect origin conservation (as performed by Nieduszynski et al, 2006) could shed light on this

issue. And one glaring omission is the authors did not attempt to construct ARS consensus sequences across the *Lachancea* lineage. How much divergence has occurred in these 10 species? Are we to assume that they all use the same consensus as *L. waltii* and *L. kluyveri*? This analysis would help distinguish the three possibilities above.

RESPONSE: We tested whether the loss of ARS activity could be due to mutations that affect the ACS by applying a motif finder program to the 4 conserved origins showing ARS activity and to the 12 lost origins used in our ARS assay in *L. thermotolerans*. An ACS was detected in all 4 conserved origins but found in only 50% of the lost origins (Supplementary Fig 15), suggesting that a fraction of the origins could be lost by mutations that affect the ACS while other losses could result from mutations that affect neighboring loci. We also searched for the ACS across the other *Lachancea* species and found a motif in 4 additional species (Supplementary Fig 14). We added a new paragraph in the main text entitled 'Origin loss and ARS activity' (p11-12), 2 supplementary figures that describe this analysis and we also now address this point in the discussion (p15).

5. Clarifications needed on figures.

Figure 5: divide into part A and part B. Why are the two graphs (in part A) not aligned when we are looking at the identical region over time?

RESPONSE: We modified Fig. 5 accordingly, dividing the figure in 2 parts and aligning the graphs in panel A.

Supplemental Figure 2: please show the MFA profiles for all chromosomes from all species potentially including them as overlays in Supplemental Figure 1.

RESPONSE: We updated the Supplementary Figure 2 by now showing all chromosomes from all species, as requested.

Supplementary Figure 7: The legend indicates that P values are included. They are missing from the figure.

RESPONSE: We are sorry for this omission. This has been corrected.

Supplementary Figure 8b: While their simulations of origin locations relative to syntenic breakpoints may not pass a significance cutoff, there does seem to be a trend to find more origins near breakpoints. Perhaps the authors could comment on this trend given other data from the literature.

RESPONSE: There is a weak association between origins and syntenic breakpoints in Supplementary Fig 10b. We calculated the probability of finding the observed number of origins near the breakpoints (at 0 to 5 genes from the breakpoints) by randomizing origin positions 1,000 times and found no significant association ($p=0.142$). However, as requested, we added a

sentence in the discussion mentioning the association between origins and breakpoints that was previously reported in the literature (p14).

Reviewers' comments:

Reviewer #1 (Remarks to the Author):

The authors have satisfactorily addressed all of my concerns.

Reviewer #2 (Remarks to the Author):

The authors have adequately addressed my questions. I think this is a very interesting paper that is now suitable for publication.

Reviewer #3 (Remarks to the Author):

Reviewer #3:

The manuscript by Nicholas Agier and colleagues is improved from the initial submission. The authors added an investigation of ARS function in the *L. waltii* genome for origins that appeared to have been lost relative to the closest sister species. They also included the ARS motifs for many of the related species. These additions add credence to many of their other conclusions. The primary data are definitely a tour-de-force and need to be in the public realm, but I have one major disagreement with the author's discussion of their data.

The major remaining issue for me is that they confuse efficiency and activation time for origins of replication—often lumping the two together as a single property. There is ample evidence in the literature that these two properties are not one and the same, regardless of what mathematical models may say. The authors even reference these studies in the introduction and then ignore them. As far as I can tell, they did no independent studies to measure either time of activation or efficiency of origin firing. What their genome wide assays illustrate is the average time of replication of different coordinates across the genome. Peaks, by definition, must correspond to origins as they accumulate increased copy number before the regions to either side. But the shape and height of peaks is influenced by the amount of passive replication from adjacent origins. As presented, neither their Trep nor MFA scans can distinguish the cause of any peak's low amplitude. It may be low because it was early firing but inefficient, or late firing, appearing to be inefficient only because an earlier firing origin is in close proximity. Since the Trep technique examined read depth at different times in S phase, I am surprised that they did not use this information to inform their origin calls. I'm guessing that, given the scatter from deep sequencing timing profiles, they did not have sufficient resolution to carry out this analysis. Throughout the results section, the authors simply accept the mathematical model and assert that origins differ in efficiency and use this categorization to discuss different classes of origins across evolutionary time. My suggestion to quickly solve this misleading interpretation is to simply define peaks of different amplitudes as "high, medium and low" and carry on with their results section referring to them as such. In the discussion, they would then be free to say what they think these categories might mean, citing the mathematical model as one interpretation. Maybe I missed it, or it is buried in supplemental material, but I did not see how they applied their mathematical model to origins from these 10 species. For that reason it seems inappropriate for them to talk about this work in the results section. My apologies if I missed it, but I only see one sentence (page 9) that states: "We used a stochastic mathematical model to infer origin firing rates from the fit of our replication timing data and derived the efficiency and firing time of each individual origin in the 10 *Lachancea* genomes from running the model.¹⁴"

Three minor issues remain for me:

1. I had asked for a better description of how an origin family is defined, and suggested a toy or

real example as an illustration. I found their additions to supplementary Figure 7b and c did not clarify the issue. I don't understand why the two parts of genome B aren't on the same horizontal axis, why origins are in the middle of genes when we know them to be in intergenes, and would like to see additional species alignments to get a sense of what "family" means. Part c doesn't help me understand what "family" means. What origin are these? Can I go back and look at the replication profiles and find them and either agree or disagree with the authors on their presence/absence? Why is LAMI present twice in the central image?

2. On page 11, they discuss the 30 ARS assays for regions that appear to have been lost from the *L. waltii* genome (based on their replication profiles). 21 did not have ARS activity. 4 were ARSs but weren't reported as chromosomal origins by Di Rienzi et al. 3 5 origins they called as lost, were in fact chromosomal origins in Di Rienzi's work. These results argue that their assays missed 5 chromosomal origins and shouldn't be considered origin losses in their experiment, but may have been below their level of detection. Therefore, I would argue that 70% (21/30) of origins were lost, not the 83% (25/30) they claim. (Please check my math and logic.) Does that mean that we can only believe 70% of the called origin losses? How does that affect the impact of their conclusions?

3. I do not understand some of their meta-analyses. I do not get the point of Figure 2c. I do not understand how conclusions are being made from Supplementary Figure 5 and the figure legend doesn't help make it clear.

Reviewers' comments:

Reviewer #1 (Remarks to the Author):

The authors have satisfactorily addressed all of my concerns.

Reviewer #2 (Remarks to the Author):

The authors have adequately addressed my questions. I think this is a very interesting paper that is now suitable for publication.

We thank Reviewers 1 and 2 for their positive appraisal of our manuscript.

Reviewer #3 (Remarks to the Author):

Reviewer #3:

The manuscript by Nicholas Agier and colleagues is improved from the initial submission. The authors added an investigation of ARS function in the *L. waltii* genome for origins that appeared to have been lost relative to the closest sister species. They also included the ARS motifs for many of the related species. These additions add credence to many of their other conclusions. The primary data are definitely a tour-de-force and need to be in the public realm, but I have one major disagreement with the author's discussion of their data.

The major remaining issue for me is that they confuse efficiency and activation time for origins of replication often lumping the two together as a single property. There is ample evidence in the literature that these two properties are not one and the same, regardless of what mathematical models may say. The authors even reference these studies in the introduction and then ignore them. As far as I can tell, they did no independent studies to measure either time of activation or efficiency of origin firing. What their genome wide assays illustrate is the average time of replication of different coordinates across the genome. Peaks, by definition, must correspond to origins as they accumulate increased copy number before the regions to either side. But the shape and height of peaks is influenced by the amount of passive replication from adjacent origins. As presented, neither their Trep nor MFA scans can distinguish the cause of any peak's low amplitude. It may be low because it was early firing but inefficient, or late firing, appearing to be inefficient only because an earlier firing origin is in close proximity. Since the Trep technique examined read depth at different times in S phase, I am surprised that they did not use this information to inform their origin calls. I'm guessing that, given the scatter from deep sequencing timing profiles, they did not have sufficient resolution to carry out

this analysis. Throughout the results section, the authors simply accept the mathematical model and assert that origins differ in efficiency and use this categorization to discuss different classes of origins across evolutionary time. My suggestion to quickly solve this misleading interpretation is to simply define peaks of different amplitudes as high, medium and low and carry on with their results section referring to them as such. In the discussion, they would then be free to say what they think these categories might mean, citing the mathematical model as one interpretation. Maybe I missed it, or it is buried in supplemental material, but I did not see how they applied their mathematical model to origins from these 10 species. For that reason it seems inappropriate for them to talk about this work in the results section. My apologies if I missed it, but I only see one sentence (page 9) that states: 'We used a stochastic mathematical model to infer origin firing rates from the fit of our replication timing data and derived the efficiency and firing time of each individual origin in the 10 *Lachancea* genomes from running the model.'

First we want to reassure the referee that we are well aware that the characteristic firing times and efficiencies are two different properties and that we do not confuse the two. However, his/her comments indicate that we did not give enough details and that some confusion in our manuscript can remain. In order to alleviate this misunderstanding we added more data to the paper (method p20 and sup figure 18) and in the next paragraph of the letter.

We think that it is not correct for the reviewer to say that what our data illustrate is only the average time of replication. Indeed, we performed entire replication kinetics by measuring the progression of DNA replication at multiple time points during S-phase for the 10 species. We then fit the experimental data with a mathematical model to infer the origin firing rates. These fits are performed by minimizing the distance between the replication timing profiles in the model and in the experimental data. We added the Supplementary Fig. 18 to illustrate this comparison between the experimental replication profiles at all time points with the fits of the theoretical model for all chromosomes in all 10 species. The results in this figure clearly show a very good agreement between the fit and the full replication timing data. From this model fit we were able to estimate directly the origin firing rates and therefore characteristic firing times (inverse of rates) a priori from the interference from nearby origins. Moreover, we estimated the origin efficiencies by direct simulation of the stochastic model with the inferred parameters as the fraction of realizations in which an origin fires before it is replicated passively. We agree with the reviewer that we did not provide such a detailed explanation of how we applied our model to the data and inferred characteristic firing time and efficiency, but only we were referring to a published work (Zhang et al., NAR, 2017). However, to clarify this issue, we now have also added a specific paragraph in the Method section (p20).

All in all and for all the reasons mentioned above, we believe that the suggestion of the reviewer to replace firing time and efficiency by peak heights would not be appropriate in our study.

Three minor issues remain for me:

1. I had asked for a better description of how an origin family is defined, and suggested a toy or real example as an illustration. I found their additions to supplementary Figure 7b and c did not clarify the issue. I don't understand why the two parts of genome B aren't on the same horizontal axis, why origins are in the middle of genes when we know them to be in intergenes, and would like to see additional species alignments to get a sense of what 'family' means. Part c doesn't help me understand what 'family' means. What origin are these? Can I go back and look at the replication profiles and find them and either agree or disagree with the authors on their presence/absence? Why is LAMI present twice in the central image?

We entirely redesigned the figure and legend in Supplementary Fig. 7b in order to explain the 3 steps we used to determine the origin families. In this new figure and legend, families of origins simply correspond to the assembly of all pairs of orthologous origins into connected components, such as those illustrated in Supplementary Fig. 7c. It could happen that 2 closely located origins in one genome get assembled into a single connected component during the aggregation of orthologous origin pairs, as shown in Supplementary Fig. 7c and now explained in the corresponding caption. We want to note that in the previous version of the manuscript we were erroneously referring to transitivity for family construction (Method page 21) which may have been confusing. We changed the corresponding text with the above explanation.

2. On page 11, they discuss the 30 ARS assays for regions that appear to have been lost from the *L. waltii* genome (based on their replication profiles). 21 did not have ARS activity. 4 were ARSs but weren't reported as chromosomal origins by Di Rienzi et al. 5 origins they called as lost, were in fact chromosomal origins in Di Rienzi's work. These results argue that their assays missed 5 chromosomal origins and shouldn't be considered origin losses in their experiment, but may have been below their level of detection. Therefore, I would argue that 70% (21/30) of origins were lost, not the 83% (25/30) they claim. (Please check my math and logic.) Does that mean that we can only believe 70% of the called origin losses? How does that affect the impact of their conclusions?

We disagree with this calculation because in this study we are considering the losses of the chromosomal activity of replication origins, regardless of their plasmid ARS activity. Therefore, both the 21 losses showing no chromosomal and no ARS activity as well as the 4 ARS showing no chromosomal activity do correspond to cases of origin losses as they are defined in our study. The correct calculation is therefore 25/30, i.e. 83% of loss validation. Note that we found a similar situation in *L. thermotolerans* where our plasmid-based ARS assay revealed that 4 out of the 12 origin losses tested retained ARS activity on plasmids. To avoid further confusion on what we call an origin loss we completed the definition presented on page 8 as follows: "What we call here an origin loss is the inactivation of a

previously active origin at the chromosomal level, regardless of its capacity to sustain the autonomous replication of a plasmid (ARS activity).”

3. I do not understand some of their meta-analyses. I do not get the point of Figure 2c. I do not understand how conclusions are being made from Supplementary Figure 5 and the figure legend doesn't help make it clear.

First, concerning Fig. 2C, we apologize for the confusion that came from the fact that we did not explain in the caption that the y axis corresponds to the normalized replication timing difference between orthologous origins. For each species, the average replication timing data (presented in Supplementary Fig. 1) were normalized between 0 and 1. The distributions of differences in normalized activation timing between orthologous origins are shown for all pairwise comparisons. In order to clarify this point we modified the legend of Fig. 2c accordingly. This figure clearly shows that the range of timing differences between orthologous origins remains between 10 and 22% whatever the phylogenetic distance between the compared species is. The absence of correlation in this graph suggests that the evolution of the replication programs did not result from a progressive change in the activation time of origins. We also added these explanations in the main text on page 7.

Concerning Supplementary Fig. 5, this figure was added in the revised version of the manuscript to answer a comment from Reviewer 2 asking (i) what happens to the correlation as it is measured at progressively larger distances from the synteny breakpoints and (ii) if there were regions of high correlation that are >25 genes away for the breakpoints. This is why we presented this figure with a very large range of windows (up to 200 genes away from breakpoints). However, we agree now with the reviewer 3 that the message conveyed by this figure was not entirely straightforward. Therefore, we decided to re-scale the figure between 5 and 40 genes because windows at more than 40 genes away from breakpoints are heavily undersampled. Local Spearman's rank coefficients are on average weaker around synteny breakpoints than within conserved synteny region (Fig. 2b). This decay can be linked to the purely technical component of profile discontinuity right at the breakpoints, i.e. in the first window of 5 genes. However, if this was the only component responsible for the correlation decay, the coefficients should increase abruptly in the next windows to reach the level found in conserved synteny regions. In contrast, the new Supplementary Fig.5 shows a gradual increase of the correlation coefficients at progressively larger distances from the breakpoints in the range 5 to 20 genes. This suggests that in addition to profile discontinuities, a biological component would also contribute to lower the correlations around breakpoints. The fusions occurring between regions with different replication timing could result in passive replication of later regions by forks from earlier regions, causing the former to replicate earlier than they otherwise would.

We added these complementary explanations in the main text on pages 6-7.

REVIEWERS' COMMENTS:

Reviewer #3 (Remarks to the Author):

I would like to thank the authors for being patient with me. I really am a fan of the work and am just striving for the best representation of it in print.

I greatly appreciate the inclusion of supplementary figure 18 and with the revised supplementary figure 7B. They will help the readers greatly.

Thank you for setting me straight on the calculation of origin loss in *L. waltii*. I now understand the calculation and agree.

I still do not understand the mathematical model used to assess origin efficiency and distinguish it from origin timing. The new section added to the methods ("Stochastic model of replication kinetics and estimate of firing rates and replication efficiency") does not help non-computational readers understand how efficiency is inferred. I was hoping for a more intuitive explanation rather than this jargon-filled description.

REVIEWERS' COMMENTS:

Reviewer #3 (Remarks to the Author):

I would like to thank the authors for being patient with me. I really am a fan of the work and am just striving for the best representation of it in print.

I greatly appreciate the inclusion of supplementary figure 18 and with the revised supplementary figure 7B. They will help the readers greatly.

*Thank you for setting me straight on the calculation of origin loss in *L. waltii*. I now understand the calculation and agree.*

I still do not understand the mathematical model used to assess origin efficiency and distinguish it from origin timing. The new section added to the methods ("Stochastic model of replication kinetics and estimate of firing rates and replication efficiency") does not help non-computational readers understand how efficiency is inferred. I was hoping for a more intuitive explanation rather than this jargon-filled description.

Response: We addressed the last concern raised by reviewer 3 by adding an explanation in the section "Modelisation of replication kinetics".